# The Inflammation Level and a Microbiological Analysis of the Anophthalmic Cavities of Unilateral Ocular Prosthesis Users: A Blind, Randomized Observational Study

**DOI:** 10.3390/antibiotics11111486

**Published:** 2022-10-27

**Authors:** Paulo Augusto Penitente, Emily Vivianne Freitas Da Silva, Marcelo Coelho Goiato, Lorena Louise Pontes Maniçoba, Victor Gustavo Balera Brito, Karina Helga Leal Túrcio, Alana Semenzin Rodrigues, Bruna Egumi Nagay, Daniela Micheline Dos Santos

**Affiliations:** 1Department of Dental Materials and Prosthodontics, Araçatuba Dental School, São Paulo State University (UNESP), Araçatuba 16015-050, São Paulo, Brazil; 2Department of Prosthodontics, School of Dentistry, University of São Paulo, São Paulo 0508-000, São Paulo, Brazil; 3Oral Oncology Center, São Paulo State University (UNESP), Araçatuba Dental School, São Paulo State University (UNESP), Araçatuba 16015-050, São Paulo, Brazil; 4Department of Basic Sciences, São Paulo State University (UNESP), School of Dentistry, Araçatuba 16015-050, São Paulo, Brazil; 5Multicenter Postgraduate Program in Physiological Sciences, Brazilian Society of Physiology, São Paulo State University (UNESP), School of Dentistry, Araçatuba 16015-050, São Paulo, Brazil; 6Department of Prosthodontics and Periodontology, Piracicaba Dental School, University of Campinas (UNICAMP), Piracicaba 13414-903, São Paulo, Brazil

**Keywords:** inflammation, eye, artificial, acrylic resins, biofilms

## Abstract

Irritation and biofilm adhesion are complaints associated with ocular prosthesis use. This study aimed to evaluate the effects of prosthesis repolishing on several conditions of anophthalmic volunteers. Participants were divided into two groups: intervention (IG, *n* = 10) and nonintervention (NIG, *n* = 6) groups. The anophthalmic cavity, contralateral eye, and prosthesis surface were evaluated at initial, day 15, and day 30 after repolishing. Microbiological analysis (colony-forming units), exfoliative cytology (conjunctiva inflammatory cells), sensory analysis (quantitative mechanical sensory test), tear production (Schirmer’s test), and conjunctival inflammation (clinical evaluation) were performed. Nonparametric tests were used to compare groups in the initial period and to analyze periods for the IG (*p* < 0.05). More microorganisms were formed in the anophthalmic socket and prosthesis than in the contralateral eye in the initial period. For IG, the anophthalmic cavity exhibited more microorganisms and inflammatory clinical signs in the initial period than at 15 and 30 after repolishing. The prosthesis showed greater accumulations of total bacteria and *Candida albicans* in the initial period than at 15 and 30 days after repolishing. The anophthalmic cavity had more palpebral inflammation than the contralateral eye. In conclusion, repolishing reduced the number of microorganisms and inflammatory signs over time.

## 1. Introduction

Ocular prostheses are intended to artificially replace the natural eye [1,2]. Through these, the remaining anatomical structures are maintained; there is physiological and functional improvement with cleaning, tear direction, and the lubrication of the anophthalmic cavity [2,3]. Acrylic resin is one of the most commonly used materials for these prostheses [1,4] because it has excellent physical and mechanical characteristics [1,5] and is biocompatible with the tissues surrounding the ocular conjunctiva [1,6,7].

However, over time, the properties of acrylic resin prostheses may change, making them more favorable to biofilm adhesion with increased surface irregularities and microcracks [7,8]. Repolishing these prostheses restores their ideal characteristics, increasing the smoothness; reduces irregularities; increases the degree of shine; and possibly mitigates biofilm formation [9,10].

Biological reactions in the anophthalmic cavity, such as conjunctival irritation and inflammation, may occur owing to ocular-prosthesis maladaptation and/or biofilm adhesion to its surface [11,12]. These factors can cause the release of harmful toxins into the conjunctiva and lead to regional inflammation [13]. Thus, prosthesis use can be uncomfortable, leading to dissatisfaction in patients [11,12,14]. Other factors that can result in inflammation include adverse environmental conditions such as dry weather and low relative humidity [15]. These factors must be considered particularly for patients with poor ocular lubrication, which is also known as dry eye [16,17]. These factors result in unfavorable clinical conditions and negative experiences for the patient, such as reduced tear production, increased conjunctival hyperemia, and irritation [18]. Contact-lens wearers are clinically influenced by environmental conditions [19,20]; therefore, prosthetic eye wearers have a high probability of developing the same disorders as contact-lens wearers.

The objective of this study was to analyze the effects of ocular-prosthesis repolishing on the microbiological, cellular, sensory, and tear production aspects of anophthalmic participants with unilateral eye prostheses before and on days 15 and 30 after repolishing. The null hypothesis tested was that ocular-prosthesis repolishing would not influence microorganism growth, inflammation, number of inflammatory cells in the anophthalmic cavity, mechanical sensory aspects, or tear production.

## 2. Results

### 2.1. Demographics and Clinical Characteristics of IG

Initially, 18 anophthalmic volunteers were selected and agreed to participate in the study. However, one volunteer withdrew because of difficulties in scheduling the analysis, and another was excluded for losing the ocular prosthesis in the initial period of the study. The remaining participants were divided into an intervention group (IG, *n* = 10), a nonintervention group (NIG, *n* = 6), and a control group (CG, *n* = 5). The mean age of the participants was 52.4 years, and most of them had a high school education (60%) and were male (60%). Trauma was the most frequent etiology (60%). Regarding their clinical history and symptoms in the last 30 days, 60% of the participants reported no dry eye, 70% had a feeling of sand in the eyes, 90% had no burning sensation in the eyes, 80% noticed secretions in the eyelashes, 90% had woken up with their eyes glued together, and 80% felt tearing. Regarding the amount of time needed to adapt to their current prostheses, 40% of the participants took up to one month to get physically used to the prosthesis, 80% took up to one month to get mentally used to the prosthesis, 60% stated that using the prosthesis slightly interfered with their daily lives, 60% stated that they thought once or twice a day about the prosthesis, and 60% felt observed on the street because of their use of the prosthesis.

### 2.2. Microbiological Analysis

In the initial period, a microbiological analysis revealed a statistical difference between the different regions analyzed in the IG compared with the CG (Figure 1) and the NIG (Figure 2) for total bacteria production (*p* < 0.001, for both), *S. aureus* and *S. epidermidis* (*p* < 0.001, for both), and *Candida albicans* (*p* < 0.001, for both). As shown in Figure 1, there was a statistically significant difference in microorganism formation in the anophthalmic socket and prosthesis in the IG. As shown in Figure 2, a statistically significantly higher number of microorganisms formed in the anophthalmic cavities and prostheses of both NIG and IG when compared with the contralateral eye.

Figure 3 shows the microbiological analysis of the anophthalmic socket and ocular prosthesis of the IG over time. The time factor negatively affected the results for total bacteria (*p* = 0.001), *S. aureus* and *S. epidermidis* (*p* = 0.008), and *Candida albicans* (*p* = 0.001) production in the anophthalmic cavity, so that the number of microorganisms was statistically larger in the initial period than on days 15 and 30. The analysis of the microorganism accumulation in the ocular prosthesis indicated interference in the total bacteria production (*p* = 0.001), with greater formation in the initial period than at 15 and 30 d, despite the time factor not having interfered in *S. aureus* and *S. epidermidis* production in the ocular prosthesis (*p* = 0.122). As for *Candida albicans* formation in the ocular prosthesis, the time factor interfered in the results (*p* < 0.001), with statistically greater accumulation in the initial period.

### 2.3. Level of Clinical Conjunctival Inflammation

The levels of palpebral conjunctiva inflammation differed between the analyzed regions in the initial period (*p* < 0.001), with larger degrees of inflammation in the anophthalmic cavity than in the contralateral eye (Figure 4).

The level of palpebral conjunctiva inflammation in the anophthalmic cavity for the IG exhibited a statistically significant difference between the analyzed periods (*p* = 0.018) (Figure 5), with a higher level of inflammation observed in the initial period than on days 15 and 30 after repolishing (*p* = 0.043, for both).

### 2.4. Conjunctival Smear Cytological Analysis

Palpebral conjunctiva cytology revealed healthy tissue in the initial period in 100% of the CG and in the contralateral eye for the IG and NIG. In the cytological analysis of the palpebral conjunctivae of the anophthalmic cavity for the IG over time, no statistically significant difference between the analyzed periods (*p* = 0.135) could be detected (Figure 6). Figure 7A–E show representative images of the cytological analysis of the conjunctivae of the anophthalmic cavity.

### 2.5. Tear Production Assessment (Schirmer’s Test)

Regarding the level of tear production in the initial period, there was no statistically significant difference in the different regions analyzed for the IG compared with the CG (*p* = 0.069). However, compared with the NIG, there was a statistically significant difference (*p* = 0.012) (Figure 8), with the contralateral eye in the IG exhibiting lower tear production than the anophthalmic cavity in the NIG (*p* = 0.034).

The results for the level of tear production in the anophthalmic cavity for the IG exhibited no interference of the time factor (*p* = 0.375).

### 2.6. Eyelid Sensitivity Assessment

An analysis of the mean force applied in the monofilament test to analyze the sensitivity in the initial period revealed no statistical differences between the different regions in the IG compared with the CG (*p* = 0.810) or the NIG (*p* = 0.662).

Mean force was applied in the monofilament test to analyze the sensitivity in the anophthalmic cavity for the IG over time. The time factor did not affect the mean force applied in the monofilament test to analyze the sensitivity in the anophthalmic cavity (upper eyelid: *p* = 0.504, lower eyelid: *p* = 0.319, or inside cavity: *p* = 0.549).

## 3. Discussion

The null hypothesis that eye-prothesis repolishing would not influence the microorganism growth and inflammation in the anophthalmic cavity was rejected, as prosthesis repolishing interfered with the results. However, the null hypothesis that eye-prothesis repolishing would not influence the number of inflammatory cells in the anophthalmic cavity, mechanical sensory aspects, or tear production was accepted.

In the IG, male participants were the most significantly affected, and trauma was the most frequent etiology (60%) for anophthalmia, in agreement with the results of Modugno et al. [21], who reported that most of the 8018 anophthalmic participants evaluated between 1927 and 2011 were male and that 63% of the participants had a traumatic etiology of anophthalmia [22]. Regarding clinical history and symptoms, most participants complained of a feeling of having sand in the eyes, secretions in the eyelashes, and a tearing sensation. Pine et al. [11] reported a high frequency of irritation in the anophthalmic cavity associated with the use of the ocular prosthesis (93%) and identified the prosthesis size/shape, polishing, manufacturing method, protein/dirt deposits on the surface, adaptation in the cavity, and cleaning method as causal factors.

The ocular prosthesis is usually made from a mold of the anophthalmic cavity and is individualized for each patient. However, there is a “dead space” between the posterior surface of the prosthesis and the anophthalmic cavity where secretion and tear residue accumulate, facilitating microorganism growth [7,23,24]. This supports the results of the present study, wherein a statistically significantly higher level of microorganism formation was observed in the anophthalmic cavity and prosthesis in the IG compared with the CG and in the anophthalmic socket and prosthesis in the NIG and IG compared with the contralateral eye for the same groups.

There was a higher level of microorganism formation in the anophthalmic cavity and ocular prosthesis compared with the contralateral eye for total bacteria, *S. aureus* and *S. epidermidis*, and *Candida albicans*. According to Arciola et al. [25], *Staphylococcus* are important prosthetic infection-related pathogens. Additionally, the anophthalmic cavity is an ideal environment for unwanted fungal proliferation—particularly *Candida albicans* yeasts [7].

The number of these microorganisms in the anophthalmic cavity for the IG was statistically larger in the initial period than that on days 15 and 30 after repolishing, indicating the effectiveness of repolishing for reducing the number of all the microorganisms evaluated. For the ocular prosthesis, there was a statistically significantly larger number of total bacteria and a larger number of *S. aureus* and *S. epidermidis* in the initial period than on days 15 and 30 after repolishing. Therefore, polishing possibly reduced the roughness and improved the surface humidity of the ocular prosthesis, hindering possible bacterial biofilm formation and optimizing the tear cleaning action [5,26].

On the basis of a literature review, Bonaque-González et al. [26] recommended that ocular prostheses be polished once a year to reduce the irritation associated with their use. Polishing increases the surface smoothness, reducing initial microorganism adhesion and the consequent bacterial colonization—particularly for bacteria such as *Staphylococcus* [7,27].

The proposed polishing protocol was effective for reducing the number of bacteria including *Candida albicans* grown on the ocular prosthesis. The aluminum oxide-based polishing paste used in the present study increased the prosthesis surface humidity, which is consistent with the results of Pine et al. [11], who reported that polishing with this paste applied with a polyurethane polishing disk reduced the amount of deposit accumulated (tear proteins, lipids, mucin, and contaminants such as microorganisms) on the ocular prosthesis. This humidity is associated with higher hydrophilicity on the surface of the acrylic resin, weakening the adherence of *Candida albicans*, which requires hydrophobic interaction with the resin base [28].

Impurity and microorganism accumulation associated with the physical presence of the prosthesis can result in surface roughness, more handling of the prosthesis over time, and increased patient irritation and discomfort [1,29]. This justified the analysis of the level of palpebral conjunctiva inflammation in the present study. In the initial period, there was greater inflammation in the anophthalmic cavity than in the contralateral eye, which was considered healthy tissue without clinical signs of inflammation, resulting from greater microorganism accumulation in the anophthalmic cavity and prosthesis compared with the contralateral eye. The analysis of the anophthalmic cavity in the IG over time indicated a higher level of inflammation in the initial period than on days 15 and 30 after polishing.

Despite this, there were no statistically significant differences between periods regarding the cytological analysis of the palpebral conjunctiva of the anophthalmic cavity in the IG. However, the percentage distribution of the cytological analysis scores changed over time, with a reduced inflammation level on day 30 after polishing (no very severe inflammation and presence of mild inflammation).

Giant papillary conjunctivitis is one of the main clinical conditions resulting from the inflammatory reaction caused by the use of ocular prostheses. It is characterized by increased mucus secretion, itching, and conjunctival irritation [13,15]. This clinical condition is characterized by a significantly increased number of inflammatory cells in the anophthalmic cavity conjunctiva—particularly neutrophils, polymorphonuclear leukocytes, mast cells, and eosinophils [16]. According to Sarac et al. [30], there is a humoral and cellular immune response involving the action of cells such as mast cells, eosinophils, neutrophils, and T lymphocytes, in addition to substances released by these cells, such as chemokines and cytokines. As a result, there is eye socket irritation and hyperemia [31].

Additionally, inflamed tissues are hypersensitive in sensitivity tests because of a lower pressure–pain threshold. In peripheral tissues, inflammation can result in voltage-gated changes in calcium and sodium channels, leading to an increased firing rate of the action potential [32]. In the present study, quantitative sensory tests identified somatosensory abnormalities and noninvasively quantified sensitivity. One of the available methods is the use of von Frey monofilaments to detect mechanical and pain sensitivity, which are considered the gold standard for this type of analysis [33,34,35]. A higher sensitivity of von Frey filaments is attributed to the central sensitization that can occur after inflammation. This sensitization may be due to pain signal processing changes in the spinal cord (148) and brain [36] caused by insufficient descending inhibitory signals [37] or excessive descending facilitatory signals [38].

In the present study, the mean force of the monofilament test in the anophthalmic cavity for the IG was numerically (but not statistically) weaker in the initial period than on days 15 and 30 after polishing, indicating a higher local sensitivity in the initial period. A reduced local sensitivity after polishing (from a numerical viewpoint) may indicate the effectiveness of repolishing for improving local inflammation.

Prosthesis polishing, which is associated with reduced microorganism growth and improved mechanical sensory aspects, was also expected to improve the tear production in the anophthalmic cavity. However, there was no statistical difference regarding the level of tear production in the anophthalmic cavity in the IG over time.

This study had two noteworthy limitations, the sample size and the short follow-up period, which was due to the COVID-19 pandemic. Further studies with a larger sample size investigating the influence of the repolishing of ocular prostheses over longer follow-up periods are encouraged.

## 4. Patients and Methods

### 4.1. Study Design

This investigation was designed as a triple-blind, randomized observational study. A software program was used to randomly allocate the volunteers into the proposed groups. It was a blind study for three operators: the one who treated the patient, the one who polished the prostheses, and the one who collected and processed the data. The study protocol was approved by the Ethics Committee of the Araçatuba Dental School of São Paulo State University (FOA/UNESP) through “Plataforma Brasil”, under opinion number 16769219.0.0000.5420. The ethical aspects were considered and approved by the committee. All volunteers were informed about the research objectives and phases and were requested to provide written informed consent. They all received a copy duly signed by the responsible researcher. This study was prepared following the STROBE guidelines [39] for observational study reports.

### 4.2. Volunteer Selection

Anophthalmic volunteers were randomly allocated into two groups with the same clinical characteristics: the NIG, which consisted of participants using ocular prostheses with no repolishing, and the IG, which consisted of participants who had their prostheses repolished. The CG included volunteers without ophthalmologic impairment or systemic diseases, for whom the analyses were standardized in the right eye.

To select the volunteers, a questionnaire containing information on the inclusion and exclusion criteria used in this study was administered (Table 1). The inclusion criteria were good general health, no systemic diseases, good cognitive ability and understanding to answer the questions, and use of an ocular prosthesis made of acrylic resin for at least two years [40]. The exclusion criteria were systemic infectious disease; chronic, acute, or subacute inflammatory/infectious disease of the anterior chamber in the anophthalmic cavity; use of systemic/local anti-inflammatory or antibiotic medication in the previous six months; use of an implant-retained prosthesis; pregnancy; smoking (due to potential mucosal irritation associated with smoke); anatomical limitations such as eyelid closure deficiency and graft; local surgical procedures in the previous six months; a history of radiotherapy treatment in the head and neck region; and neuropathic pain [40].

All analyses were performed in the anophthalmic socket and in the contralateral eye for both anophthalmic groups (IG and NIG) and the CG in the initial period and on days 15 and 30 after ocular-prosthesis repolishing for the IG.

### 4.3. Demographic Data Collection and Psychosocial Profile

Sociodemographic data (sex, marital status, educational level) were collected [41]. In addition, a form was used to assess the psychosocial profiles, as proposed by Nicodemo and Ferreira [42] and by Goiato et al. [40]. The form consisted of 43 questions (closed, semi-open, open, and Likert scale) divided into five blocks. This analysis was performed by a single professional in a comfortable and quiet place.

The form included questions related to clinical history and symptoms (adapted from the study of Ghislandi and Lima [22]). In addition, the etiologies of eye loss were categorized into trauma, congenital origin, and pathologies [21].

Questions were also administered to analyze the time taken to adapt to prosthesis use, as proposed by Rasmussen et al. [43], to assess the patient’s adaptation to rehabilitation.

### 4.4. Microbiological Analysis of Anophthalmic Cavity and Ocular Prosthesis

For microbiological analysis, periorbital tissue antisepsis was initially performed with 2% chlorhexidine gluconate degerming agent to remove dirt and avoid sample contamination. Three sterile rayon swabs (Rayswab, Difco, Oakville, ON, Canada) were used to collect conjunctival secretion from the anophthalmic cavity and material from the ocular prosthesis and contralateral eye. The swabs were soaked in a sterile saline solution (0.9% sodium chloride) in their respective tubes [10,23].

Initially, the prosthesis was removed with sterile gauze, and the first swab was wiped over the internal and external surfaces of the ocular prosthesis. The second swab was wiped on two different regions of the anophthalmic conjunctiva (inferior and superior fornix fundus regions), and the third swab was wiped over the inner part of the lower eyelid of the contralateral eye [10,23].

Then, the samples were transported on ice to a laminar flow chamber (sterile environment), where they were plated in three different culture media. Samples were prepared by using 10 mL of 0.9% NaCl and vortexed for 1 min [44]. One drop of 20 μL of each sample was plated on a blood agar (BHI + blood) medium for the total bacteria culture, a salted mannitol medium for the Staphylococcus species (S. epidermidis and S. aureus), and a Sabouraud Dextrose Agar medium with chloramphenicol for the Candida albicans culture [44]. The plates were incubated at 37 °C for 48 h in an aerobic environment [44]. Subsequently, the colony-forming units (CFU) were counted using a stereoscopic magnifying glass, and the data were expressed as CFU per milliliter [44].

The number of microorganisms throughout the study period was assessed for the anophthalmic socket and ocular prosthesis since these regions were directly affected by the polishing procedure.

### 4.5. Clinical Analysis of Degree of Inflammation

The palpebral conjunctivae were classified via the inflammation scale proposed by Pine et al. [45] according to clinical signs of vasodilation, edema, and apparent roughness on the conjunctival surface—particularly the 10 mm wide region on the lower palpebral conjunctiva adjacent to the eyelid margin. The levels of this scale are as follows: 0 (less severe with no inflammatory signs, smooth and satiny surface); 1 (discreet onset of reddish regions of <1 mm at the eyelid margin); 2 (a few reddish, uniform, and limited papillary regions); 3 (nonuniform appearance with more reddened and vasodilated regions); and 4 (highest inflammatory level, the most severe, with a giant papillary aspect and nonuniform edema) [45]. For this, the volunteers’ lower eyelids were everted with the aid of a sterile swab and classified according to the clinical signs of inflammation [45].

### 4.6. Analysis of Inflammatory Cells through Conjunctival Exfoliative Cytology

Inflammatory cells were analyzed through exfoliative cytology, with the lower and upper tarsal anophthalmic cavity conjunctiva being collected using a sterile swab. Then, the collected material was used to prepare two histological slides with unidirectional smear in rotational movement. Subsequently, the slides were fixed in absolute alcohol and stained with hematoxylin and eosin [39,46]. The slides were then examined using a binocular microscope (Axio Scope.A; Carl Zeiss; Munich, Germany) to determine cell types and morphology [39]. Thus, it was possible to qualify the inflammatory infiltrate according to the cell types, such as neutrophils, eosinophils, mast cells, basophils, and lymphocytes, in relation to epithelial cells [39,46,47,48].

The conjunctiva of the lower eyelid of the contralateral eye and of the CG were evaluated in the initial period and on days 15 and 30 after ocular-prosthesis repolishing in the IG.

The following categories were used in the cytological analysis [47,49]: (A) Score 0—100% epithelial cells, absence of mucin, classified as healthy tissue; (B) Score 1—most epithelial cells, with few spaced inflammatory cells (neutrophils), absence of other types of inflammatory cells, absence of mucin, classified as tissue with initial and mild inflammation; (C) Score 2—large number of inflammatory cells in relation to epithelial cells, but with absence of mucin and little or no degraded cells, normal nucleus/cytoplasm ratio, mild to moderate tissue inflammation; (D) Score 3—high inflammatory infiltrate with a majority of neutrophils, followed by lymphocytes and eosinophils, small number of epithelial cells detected, little spacing between cells and large amount of mucin, presence of phagocytosed cells and reduced nucleus/cytoplasm ratio, moderate to high tissue inflammation; (E) Score 4—high inflammatory infiltrate, undetected epithelial cells, predominance of neutrophils, followed by lymphocytes and few eosinophils, large amount of mucin, presence of phagocytosed cells and reduced nucleus/cytoplasm ratio, high tissue inflammation [33,36]. Representative images were obtained at 400× magnification.

### 4.7. Tear Production Analysis (Schirmer’s Test)

The tear production was measured using Schirmer’s test strips (Tear Strips, AIVIMED GmbH, Wiesbaden, Germany). The strip was placed on the tear meniscus of the lower eyelid on both sides (anophthalmic cavity and contralateral eye) for the IG and NIG and in the right eye for the CG for 5 min [50,51]. This test quantified the tear production according to the length of absorbed tape, with a length of <5 mm indicating an abnormality suggestive of dry eye disease and low tear production rate [22,51].

### 4.8. Mechanical Sensitivity Assessment

A mechanical sensory test was performed using nylon monofilaments (Touch-Test Sensory Evaluators, North Coast Medical Inc., Morgan Hill, CA, USA) calibrated to exert specific forces ranging from 0.008 to 300 g/mm^2^ when flexed, according to the diameter used [52]. Initially, the volunteers removed their prostheses and waited a minimum period of 5 min to start the measurements [17]. The volunteers were asked to close their eyes and raise their left hands when noticing the stimulus.

The analysis was performed by positioning the monofilament vertically and perpendicularly to the external periorbital points; the superior and inferior mesial, central, and distal eyelid of the contralateral eye and anophthalmic cavity; and an internal point of the anophthalmic cavity for the IG and NIG and the external eyelid of the right eye for the CG.

For this, light pressure was applied for approximately 2 s on each point until the monofilament folded [32,53,54,55]. Monofilaments were tested in sequential order, from smallest to largest. The smallest-gauge monofilament recognized by the volunteer (mechanical detection threshold) was recorded. If the volunteer did not recognize the stimulus, a new monofilament of a larger caliber was used in the interval between attempts.

The mean obtained for the eyelids of each volunteer was calculated. In addition, a point on the palmar surface of the thenar muscle, which represented a region distant from the trigeminal area and not directly related to it, was used as a reference and parameter region to evaluate the volunteers’ general sensitivity threshold [56].

### 4.9. Ocular-Prosthesis Repolishing

The evaluated ocular prostheses underwent all the finishing and polishing protocols during manufacture before installation, as described by Goiato et al. [5].

The repolishing process used in the present study for the ocular prostheses of the IG was based on the protocol proposed by Barreto et al. [57], which uses universal aluminum oxide polishing paste (Kota Indústria e Comércio LTDA; São Paulo, SP, Brazil) and a felt wheel (Shofu Inc; Kyoto, Japan) coupled to a lathe motor (Nova OGP Indústria e Comércio LTDA; Bragança Paulista, SP, Brazil) operating at a speed of 3000 rpm for 5 s on each side. Repolishing was performed by the same operator and aimed to provide a better finish and increased smoothness of the prosthesis, mimicking a natural appearance [42,58,59].

After all the analyses, the prostheses of the NIG group were identically repolished to avoid differences in the quality of care provided to the participants.

### 4.10. Statistical Analysis

Statistical analysis was performed using SPSS software (version 24.0, SPSS Inc., Chicago, IL, USA). Psychosocial profile data were subjected to descriptive statistics. The data were subjected to a normality analysis using the Kolmogorov–Smirnov test, with no normality of the microbiology data, level of sensitivity, tear production, level of inflammation, or cytology. The Kruskal–Wallis test was used to compare the groups with regard to the microorganism proliferation, mean force applied in the monofilament test for analyzing the sensitivity, and tear production level in the initial period. The Friedman test was used to analyze the microorganism proliferation, sensitivity level, and tear production level in the IG over time. After the descriptive statistics, a comparative analysis was performed between regions in the initial period, followed by the Mann–Whitney test to analyze the inflammation level of the palpebral conjunctivae of the IG. The Friedman test was used for a comparative analysis over time of the anophthalmic cavity. Regarding the cytology of the IG, after descriptive statistics, the Cochran Q test was used for a comparative analysis of the anophthalmic cavity over time. A significance level of 5% was considered in all the analyses.

## 5. Conclusions

Ocular-prosthesis repolishing reduced the number of CFUs during a 30 days follow-up period after the procedure. In addition, there was a statistically significant reduction in clinical inflammation signs in the anophthalmic conjunctivae. Thus, ocular-prosthesis repolishing is essential for microbial and inflammatory control in the anophthalmic cavity.

## Figures and Tables

**Figure 1 antibiotics-11-01486-f001:**
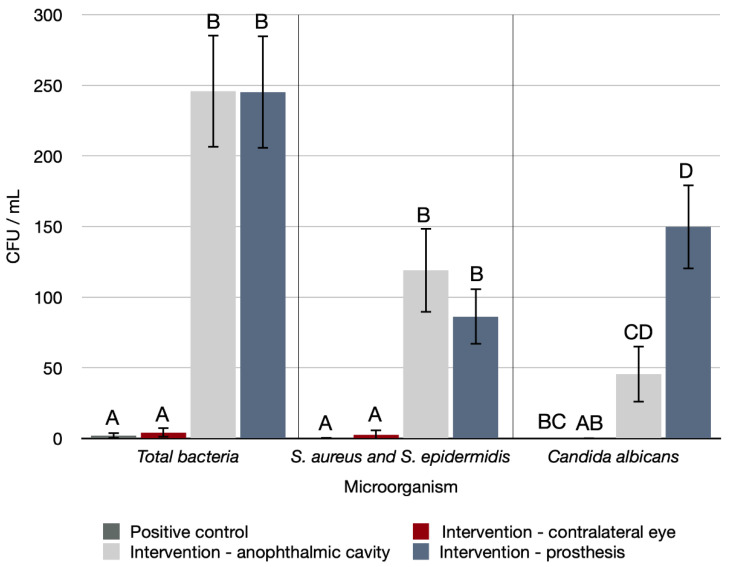
Number of CFUs/mL of the different types of microorganisms evaluated in the initial period in the different regions analyzed in the IG compared with the CG. Capital letters indicate a statistically significant difference for each type of microorganism separately (Kruskal–Wallis, *p* < 0.05).

**Figure 2 antibiotics-11-01486-f002:**
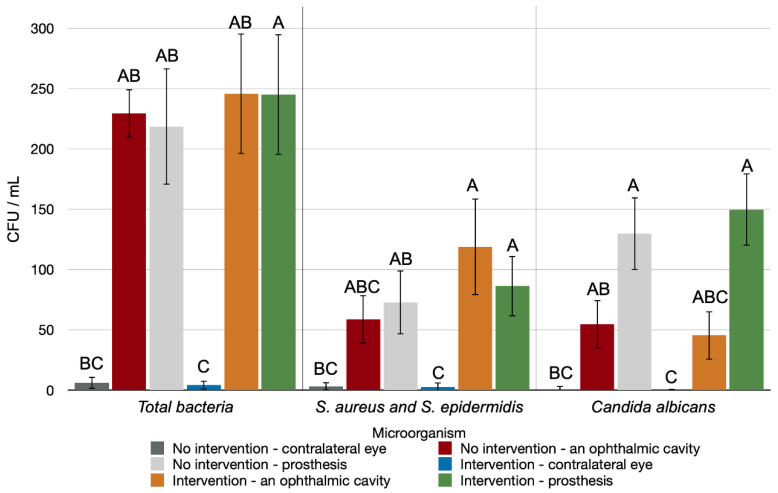
Number of CFUs/mL of the different types of microorganisms evaluated in the initial period in the different regions analyzed in the IG compared with the NIG. Capital letters indicate a statistically significant difference for each type of microorganism separately (Kruskal–Wallis, *p* < 0.05).

**Figure 3 antibiotics-11-01486-f003:**
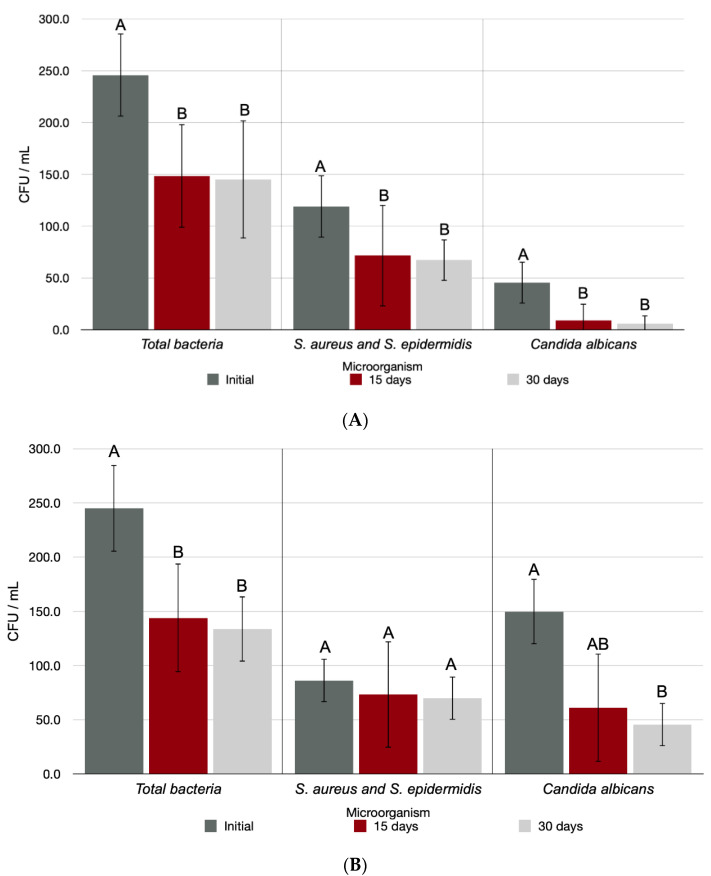
Number of CFUs/mL of the different types of microorganisms evaluated in the anophthalmic socket (**A**) and in the ocular prosthesis (**B**) of the IG over time. Capital letters indicate a statistically significant difference for each type of microorganism separately (Friedman, *p* < 0.05).

**Figure 4 antibiotics-11-01486-f004:**
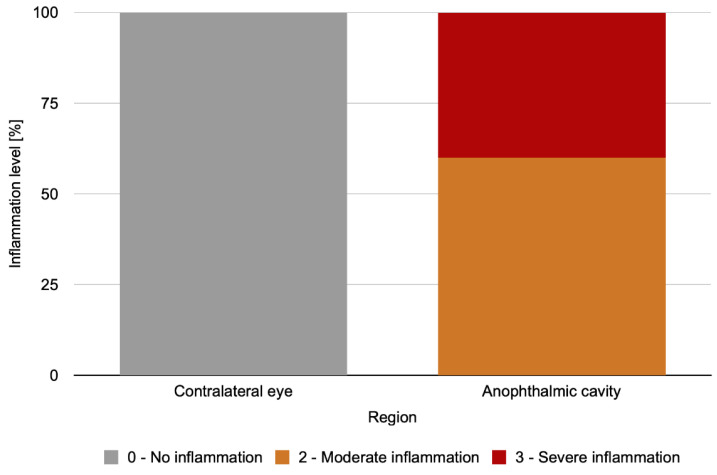
Percentage distribution of the level of palpebral conjunctiva inflammation in the initial period in the different regions analyzed in the IG.

**Figure 5 antibiotics-11-01486-f005:**
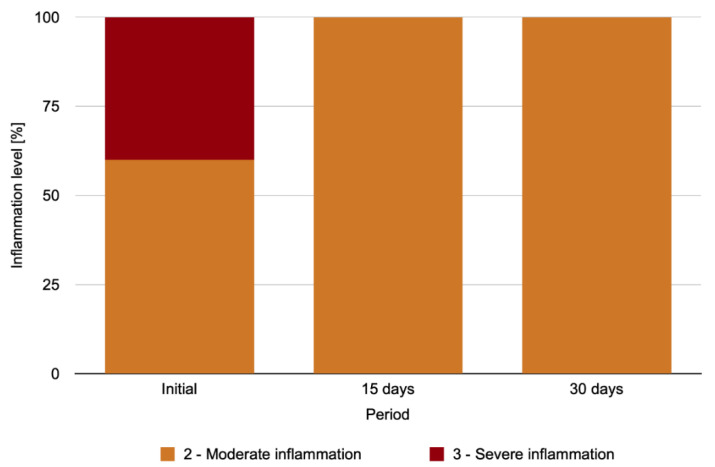
Percent distribution of the level of palpebral conjunctiva inflammation in the anophthalmic socket over time in the IG.

**Figure 6 antibiotics-11-01486-f006:**
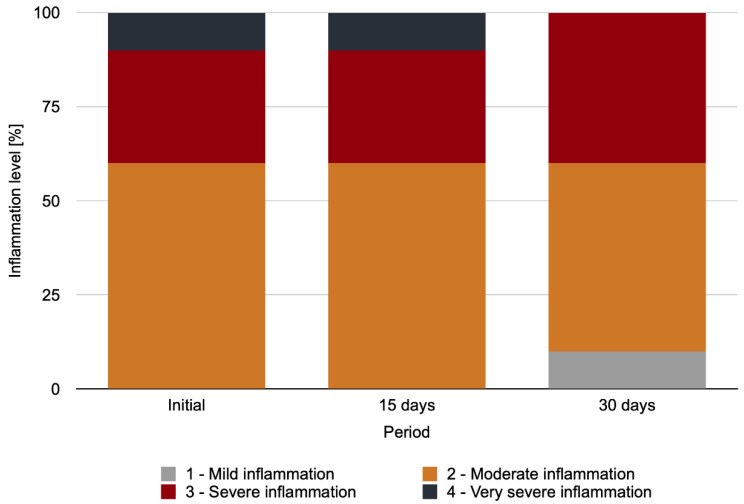
Percentage distribution of cytological analysis scores for the palpebral conjunctivae of the anophthalmic cavity over time in the IG.

**Figure 7 antibiotics-11-01486-f007:**
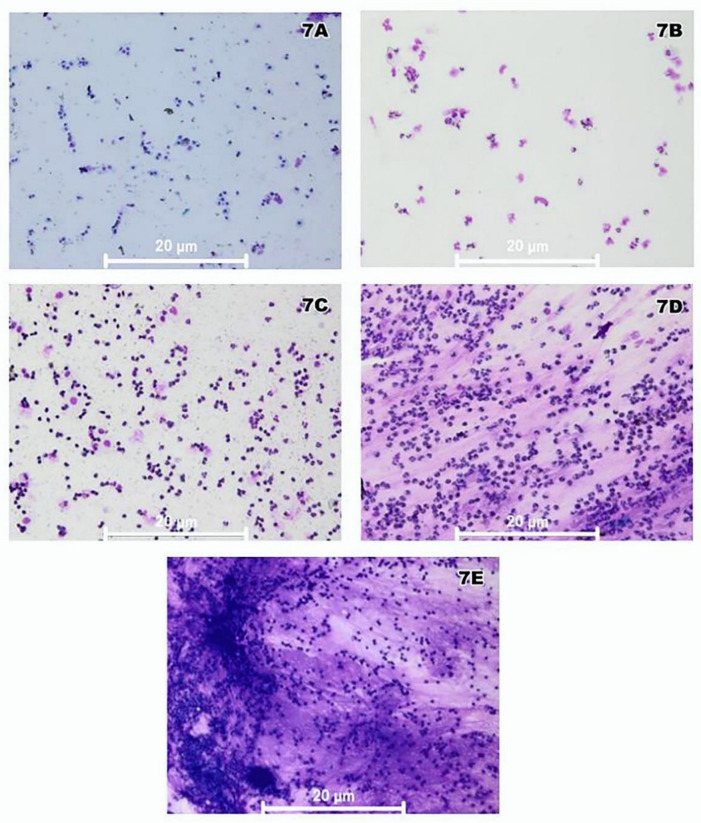
Representative images of the cytological analysis of the conjunctivae under an optical microscope at 400× magnification. (**A**) Score 0; (**B**) Score 1; (**C**) Score 2; (**D**) Score 3; (**E**) Score 4.

**Figure 8 antibiotics-11-01486-f008:**
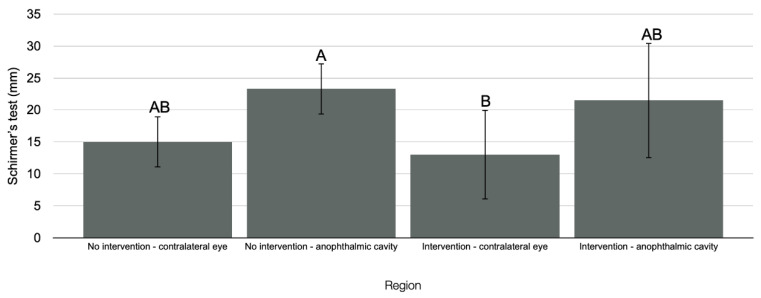
Tear production level in the initial period in the different regions analyzed in the IG compared with the NIG. Capital letters indicate a statistically significant difference (Kruskal–Wallis, *p* < 0.05).

**Table 1 antibiotics-11-01486-t001:** DN4 questionnaire administered during the process of including/excluding volunteers in/from the study.

Please complete the questionnaire by marking one answer for each number of the four questions below: VOLUNTEER’S INTERVIEW
Question 1: Does your pain have one or more of the following characteristics? 1- Burning YES ( ) NO ( ) 2- Painful cold sensation YES ( ) NO ( ) 3- Electric shock YES ( ) NO ( )
Question 2: Is one or more of the following symptoms present in the same area as your pain? 4- Tingling YES ( ) NO ( ) 5- Pins-and-needles feeling YES ( ) NO ( ) 6- Numbness YES ( ) NO ( ) 7- Itching YES ( ) NO ( )
PATIENT EXAMINATION Question 3: Is the pain located in an area where physical examination may reveal one or more of the following characteristics? 8- Hypoesthesia on touch YES ( ) NO ( ) 9- Needle pricking hypoesthesia YES ( ) NO ( )
Question 4: In the painful area, pain may be caused or increased by: 10- Brushing YES ( ) NO ( ) SCORE 0—For each negative item 1—For each positive item Neuropathic pain: Total score from 4/10

## Data Availability

Not applicable.

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
