# Peer review of "The Inflammation Level and a Microbiological Analysis of the Anophthalmic Cavities of Unilateral Ocular Prosthesis Users: A Blind, Randomized Observational Study"

_antibiotics, 2022, doi:10.3390/antibiotics11111486_

Round 1

Reviewer 1 Report

An interesting and complex article.

A wide series of determinations: microbiological analysis, exfoliative cytology, sensory analysis (quantitative mechanical sensory test), tear production, and conjunctival inflammation were performed to determine the effects of ocular prosthesis repolishing.

Regarding microbiological analysis, how were the microbial strains identified? Were only these three strains present?

In the discussion part it is postulated that polishing the ocular prosthesis could reduce biofilm formation. Was the presence of biofilm identified in the study patients?

Figures 1-3 are difficult to understand. The colors used in figures 1 and 3 have close shades, it is difficult to interpret. also those capital letters used to mark the statistical differences that appear are hard to understand.

Figure 3 shows only 2 regions: anophthalmic socket and ocular prosthesis, regions where the evolution of the number of microorganisms was followed over time. This fact is not reflected in the material and method section where the collection procedure from three distinct regions is described and it is understood that the same procedure was followed throughout the study period. The material and method section must be completed to specify that the collection from three different regions was only done at the beginning of the study.

However, the study has some limitations: the small number of patients in the study and the short follow-up period.

Author Response

RESPONSE TO REVIEWER 1

We thank the reviewer for their comment.

  1. Reviewer: Regarding microbiological analysis, how were the microbial strains identified? Were only these three strains present?

Response: We accessed the formation of total bacteria culture, the Staphylococcus species (S. epidermidis and S. aureus), and the Candida albicans culture. The total bacteria culture was accessed to evaluate the total amount of bacteria formed. However, according to Arciola et al. (Arciola, C.R.; Campoccia, D.; Speziale, P.; Montanaro, L.; Costerton, J.W. Biofilm formation in Staphylococcus implant infections: a review of molecular mechanisms and implications for biofilm‐resistant materials. Biomaterials. 2012, 33, 5967‐5982), Staphylococcus are important prosthetic infection-related pathogens. Therefore, the Staphylococcus species were accessed separately. In addition, according to Andreotti et al. (Andreotti, A.M.; Sousa, C.A.; Goiato, M.C.; Silva, E.V.F.D.; Duque, C.; Moreno, A.; Santoso, D.M.D. In vitro evaluation of microbial adhesion on the different surface roughness of acrylic resin specific for ocular prosthesis. Eur J Dent. 2018, 12, 176-183), the anophthalmic cavity is an ideal environment for unwanted fungal proliferation — particularly Candida albicans yeasts. This explains the microbial strains selected in the present study. They were identified by using selective and differential medium in which the sample was plated: a blood agar (BHI + blood) medium for the total bacteria culture (all species present in the biofilm), a mannitol salt agar (MSA) for the Staphylococcus species (S. epidermidis and S. aureus), and a Sabouraud Dextrose Agar (SDA) medium with chloramphenicol for the Candida albicans culture. It is important to emphasize that each selective medium has unique composition that allow the isolation of a particular bacteria species or genus. For example, because MSA has a high concentration of salt, it is recommended for the isolation of pathogenic Staphylococci, since such bacteria have the unique ability of growing on a high salt containing media. In the case of SDA, this medium is recommended for the fungal cultivation, since it is formulated with a low pH (5.6) to slightly inhibit bacterial growth; thus, we supplemented the media with antibiotic (chloramphenicol) to totally inhibit bacteria. Regarding the total microbial growth, we used a well-known non-selective media supplemented with blood to provide a better environment (i.e. more growth factors and nutrients) for the growth of the total microorganisms.

  1. Reviewer: In the discussion part it is postulated that polishing the ocular prosthesis could reduce biofilm formation. Was the presence of biofilm identified in the study patients?

Response: No, we did not identify the presence of biofilm. Thus, we removed the sentence from the discussion section (Page 12, lines 250-252).

  1. Reviewer: Figures 1-3 are difficult to understand. The colors used in figures 1 and 3 have close shades, it is difficult to interpret. Also those capital letters used to mark the statistical differences that appear are hard to understand

Response: Alterations were made in the Graphs and we are willing to make further modifications, if needed. S. epidermis and S. aureus were not grouped. We assessed the Staphylococcus species. We used mannitol medium for the Staphylococcus species (S. epidermis and S. aureus). We added vertical lines in the graph to separate Total bacteria, Staphylococcus species, and Candida albicans aiming a better understanding. Capital letters indicate a statistically significant difference for each type of microorganism separately (Kruskal–Wallis, p < 0.05).

We changed the column colors to facilitate interpretation and increased the capital letters that indicate statistical differences.

  1. Reviewer: Figure 3 - Figure 3 shows only 2 regions: anophthalmic socket and ocular prosthesis, regions were the evolution of the number of microorganisms was followed over time. This fact is not reflected in the material and method section where the collection procedure from three distinct regions is described and it is understood that the same procedure was followed throughout the study period. The material and method section must be completed to specify that the collection from three different regions was only done at the beginning of the study.

Response: We apologize for the misunderstanding. The number of microorganisms throughout the study period was assessed for the anophthalmic socket and ocular prosthesis since these regions were directly affected by the polishing procedure. We adjusted the Material and Method section accordingly. 

  1. Reviewer: However, the study has some limitations: the small number of patients in the study and the short follow-up period.

Response: A paragraph describing the limitation of the study (‘This study had two noteworthy limitations such as the sample size and the short follow-up period, due to the COVID-2019 pandemic. Further studies with a larger sample size investigating the influence of repolishing of ocular prostheses on longer follow-up periods are encouraged) was added to the manuscript, as suggested by the reviewer.

Reviewer 2 Report

Dear authors, I did enjoy reading your manuscript. The findings presented are really interesting. I have however some suggestions as to improve especially the microbiological part (being a microbiologist). There are some crude errors in this regard which I allowed myself to point out in the PDF file using the comments function of adobe PDF. Also the graphs need some more work to be better understandable from a readers point of view, also this I commented on in the PDF.

Kind regard!

Author Response

RESPONSE TO REVIEWER 2

We thank the reviewer for their comment.

  1. Reviewer: Page 2, line 48 | ‘release of’

Response: ‘release harmful’ was change ‘release of harmful’

  1. Reviewer: Page 2, line 57 | ‘>those in< can be left out of this sentence, it is not necessary for the contexto.

Response: ‘those in” was removed

  1. Reviewer: Page 2, line 72-74 | ‘Has this been related to a time frame? in the last month, ever, at the moment of filling in the questionair/during the talk with the physician? This would be interesting to add here, please.’

Response: In the text was added the time ‘in the last thirty days’.

  1. Reviewer: Figures 1, 2 and 3 - I dont understand this graph. First, why did you group S. epidermidis and S. aureus? Secondly, species names need to be italic. Most importantly I cannot follow the use of the capital letters for indicating the statistical difference - what do they stand for and does for example the a above the first two bars of total bacteria mean that those are signigicantely different? Please clarify.

Response: Alterations were made in the Graph and we are willing to make further modifications, If needed. S. epidermis and S. aureus were not grouped. We assessed the Staphylococcus species. We used mannitol medium for the Staphylococcus species (S. epidermis and S. aureus). We added vertical lines in the graph to separate Total bacteria, Staphylococcus species, and Candida albicans aiming a better understanding. Capital letters indicate a statistically significant difference for each type of microorganism separately (Kruskal–Wallis, p < 0.05). We changed the column colors to facilitate interpretation and increased the capital letters that indicate statistical differences.

  1. Reviewer: Figure 4, 5 and 6 - If you give this in % you should add this to the axis description: Inflammation level [%]. The color red does not appear in the graph? The color orange does not appear in the graph? Just a suggestion for the color scheme you are using: it might be more logical to use more intense colors for more severe inflammation? It does seem counterintuitive for mild inflammation to be dark red while severe inflammation is bluish and very severe is orange....

Response: We corrected the axis description and we removed legends that were not presented in the results. In addition, we changed the color scheme, as suggested by the reviewer.

  1. Reviewer: Page 6, line 150-151 | ‘there was no statistically significant difference between the analyzed periods (p = 0.135) (Figure 6)’ Might be better to rewrite e.g.: ..., no statistically significant difference between the analyzed periods (p=0.135) could be detected (Figure 6).

Response: We corrected in the text, as suggested by the reviewer.

  1. Reviewer: Figure 7 - Are you sure it is not 400x? Also what lenghts are the scale bars?

Response: We checked and we used a 400x-magnification. The lengths are 20 um, as altered in the Figure.

  1. Reviewer: Figure 8 - mm is not a measurement for a volume. I am not sure that this figure is needed.

Response: Figure 8 was removed.

  1. Reviewer: Figure 9 - again see comments above regarding the capital letters in the graphs, and also here: mm is not a unit used for a volume.

Response: Capital letters indicate a statistically significant difference (Kruskal–Wallis, p < 0.05). The axis description was changed.  It refers to the length of absorbed tape in the Schirmer’s Test.

  1. Reviewer: Figure 10 - where is the time indicated in the graph? I am also not sure if this graph adds to the information given in the text, I think it can be left out.

Response: We apologize for the mistake but the Figure 10 was incorrect. However, since no statistical difference was observed, the figure was removed.

  1. Reviewer: Figures 11, 12 and 13 - Both figures do not add to the information given already in the text and might be left out of the manuscript. Again, no additional information is given in the gaph.

Response: Figures 11 to 13 were removed.

  1. Reviewer: Page 12, line 209 - 212 | ‘The null hypothesis that eye-prothesis repolishing would not influence the microorganism growth, inflammation, number of inflammatory cells in the anophthalmic cavity, mechanical sensory aspects, or tear production was rejected, as prosthesis repolishing interfered with the results.’

It did not interfere with the results. It showed a difference in MO development.

Response: That sentence has been rewritten.

  1. Reviewer: Page 12, line 234-236 | ‘S. aureus preferentially lives on mucosalsurfaces [26], and S. epidermidis adheres to polymer surfaces [27] such as acrylic resin ocular prostheses.’ This is simply not correct. S. aureus is known to be a very good biofilm forming species (especially clinical relevant strains exhibit this feature as on of their pathological features) and are the most common bacterial species found on prosthesis materials. S. epidermidis is also found on prosthesis, that is true but a lot less frequent than S. aureus as S. epidermidis are not a strictly pathogenic species but rather common skin bacteria.

Response: The sentence was removed.

  1. Reviewer: Page 12, line 244 | did you evaluate the biofilm formation? From the data reported here you did simply measure the bacterial count, that cannot directly be related to biofilm formation. I agree that this is a likely scenario, but as you did not perform a biofilm evaluation you cannot state this as a fact here.

Response: The sentence was rewrite to ‘Therefore, polishing possibly reduced the roughness and improved the surface humidity of the ocular-prosthesis, hindering possible bacterial biofilm formation and optimizing the tear cleaning action. [5,28].

  1. Reviewer: Page 12, lines 250-252 | ‘After the initial adhesion of bacteria, extracellular polymers are produced, and biofilm is formed, with increased local acidity, strengthening the adhesion of Candida albicans to the acrylic resin”.

Again, you did not show that biofilms developed, you only have the CFU...

Response: This sentence was removed.

  1. Reviewer: Page 12, lines 250-252 | ‘The latter is a large fungus (46 µm) [32] with long filaments. [33]’

This is not correct. Candida albicans yeast cells are about 5 to 7µm tall. This yeast can form pseudohyphea which can grow rather long (surely also 46µm) but a single cell is not that big. https://www.sciencedirect.com/science/article/pii/S0966842X04001180

Response: This sentence was removed.

  1. Reviewer: Page 13, line 283 | There is a different type of reference, please correct.

Response: The reference was corrected.

  1. Reviewer: Page 15, line 363 | material rather than biofilm.

Response: Ok.

  1. Reviewer: Page 15, lines 372-373 | ‘For a fungal analysis, Sabouraud Dextrose Agar medium with chloramphenicol was used for the Candida albicans culture.’ Maybe rephrase to: Sabouraud Dextrose Agar medium with chloramphenicol was used for the Candida albicans culture.

Response: This sentence was rephrased, as suggested by the reviewer.

  1. Reviewer: Page 15, lines 376 | Please check your reference style, in the top part of the manuscript you give the reference after the ".". Here beforethe "."

Response: Ok, references checked.

  1. Reviewer: Page 15, lines 376 | ‘bacteriological oven’ - this is called an incubator.

Response: The term bacteriological oven was changed.

  1. Reviewer: Page 15, lines 377-378 | ‘Subsequently, the colony-forming units (CFUs) per milliliter (mL) were counted using a stereoscopic magnifying glass.’ Please specifiy: yhow much of the sodium chloride solution was plated? you counted the colonies oon the plates and calculated the amount per mL i guess? Then you did not count the CFU´s per mL directly. This needs to be rephrased to be methodologically clear.

Response: ‘Samples were prepared by using 10 mL of 0.9% NaCl and vortexed for 1 min. One drop of 20uL of each sample were plated on a blood agar (BHI + blood) medium for the total bacteria culture, a salted mannitol medium for the Staphylococcus species (S. epidermidis and S. aureus), and a Sabouraud Dextrose Agar medium with chloramphenicol for the Candida albicans culture. The plates were incubated at 37°C for 48 h in an aerobic environment. Subsequently, the colony-forming units (CFU) were counted using a stereoscopic magnifying glass, and the data were expressed as CFU per milliliter.’ This information was added in the manuscript with reference (45. Nagay, B.E.; Dini, C.; Cordeiro, J.M.; Ricomini-Filho, A.P.; de Avila, E.D.; Rangel, E.C.; da Cruz, N.C.; Barão, V.A.R. Visible-light-induced photocatalytic and antibacterial activity of TiO2 codoped with nitrogen and bismuth: new perspectives to control implant-biofilm-related diseases. ACS Appl Mater Interfaces. 2019, 11, 18186-18202).

  1. Reviewer: Page 16, lines 440 | ‘a new monofilament of a larger caliber was used in the interval between attempts.’

This is very confusing- you used a larger caliber between attemps and then switched back to the smaller one. Or did you change to a larger caliber in the intervall between the attempts and then used this larger one?

Response: Monofilaments were tested in sequential order, from smallest to largest. This information was added in the manuscript.